# SNAI1-Driven Sequential EMT Changes Attributed by Selective Chromatin Enrichment of RAD21 and GRHL2

**DOI:** 10.3390/cancers12051140

**Published:** 2020-05-02

**Authors:** Vignesh Sundararajan, Ming Tan, Tuan Zea Tan, Qing You Pang, Jieru Ye, Vin Yee Chung, Ruby Yun-Ju Huang

**Affiliations:** 1Cancer Science Institute of Singapore, National University of Singapore, Center for Translational Medicine, Singapore 117599, Singapore; csivsun@nus.edu.sg (V.S.); tmluanbu1989@163.com (M.T.); csittz@nus.edu.sg (T.Z.T.); pqy@u.nus.edu (Q.Y.P.); vinyeechung@gmail.com (V.Y.C.); 2Department of Obstetrics and Gynaecology, Yong Loo Lin School of Medicine, National University of Singapore, Singapore 119077, Singapore; 3School of Medicine & Graduate Institute of Oncology, College of Medicine, National Taiwan University, Taipei 10051, Taiwan; jieruye@ntu.edu.tw

**Keywords:** EMT spectrum, cohesin, RAD21, GRHL2, chromatin looping, PERP, ERBB3

## Abstract

Over two decades of research on cancer-associated epithelial-mesenchymal transition (EMT) led us to ascertain the occurrence of transitional intermediate states (collectively referred to as the EMT spectrum). Among the molecular factors that drive EMT, SNAI1 plays an indispensable role in regulating other core transcription factors, and this regulation is highly context-dependent. However, molecular investigation on this context-dependent regulation is still lacking. Using two ovarian cancer cell lines, we show that SNAI1 regulation on other core EMT-TFs switches from a repressive control in highly epithelial cells to an activation signaling in intermediate epithelial cells. Upon further scrutiny, we identify that the expression of early epithelial genes *PERP* and *ERBB3* are differentially regulated in SNAI1-induced sequential EMT changes. Mechanistically, we show that changes in *PERP* and *ERBB3* transcript levels could be correlated to the selective enrichment loss of RAD21, a cohesin component, at the distal enhancer sites of *PERP* and *ERBB3*, which precedes that of the proximal promoter-associated sites. Furthermore, the RAD21 enrichment at the distal enhancer sites is dependent on GRHL2 expression. In a nutshell, the alteration of GRHL2-associated RAD21 enrichment in epithelial genes is crucial to redefine the transition of cellular states along the EMT spectrum.

## 1. Introduction

Metastasis remains the most devastating and enigmatic phase of cancer progression, which accounts for more than 90% of patient mortality. Among several mechanisms that expedite metastasis, the pivotal role of epithelial-to-mesenchymal transition (EMT) is becoming increasingly recognized. In a nutshell, EMT is a cellular reprogramming process that enables epithelial cells to lose their key features, including intercellular adhesion junctions, to acquire mesenchymal characteristics such as cell motility and front-rear polarity [1]. EMT is accomplished through complex coordination between signaling pathways and the underlying transcriptional regulation of proteins that are critical in maintaining epithelial integrity [2,3]. Rather than a simple binary shift from an epithelial to a mesenchymal phenotype, EMT occurs as a continuum of sequential, transient dedifferentiation processes with several intermediate/transitional phenotypes [4]. During the early phases of EMT, epithelial cells are often prone to losing their epithelial characteristics, such as the disassembly of cell-cell junctions (tight junctions, adherens junctions, and desmosomes), resulting in the prominent loss of the epithelial marker, E-cadherin, without developing any traces of mesenchymal traits [5]. Subsequently, these early EMT cells further transition into multiple, intermediary cellular states with varying levels of epithelial and mesenchymal marker expressions. Each distinct intermediary phenotype has distinct morphological, transcriptional, and epigenetic features [6,7]. In particular, using a panel of ovarian cancer cell lines, we have identified four epithelial-mesenchymal subgroups: epithelial (E), intermediate epithelial (IE), intermediate mesenchymal (IM), and mesenchymal (M) collectively referred to as the EMT spectrum [8]. Studies focusing on tumor molecular profiling reveal the presence of intermediate EMT cell states, and these subpopulations are often correlated with accelerated tumor aggressiveness and therapeutic resistance [9,10,11,12,13]. Finally, the total acquisition of mesenchymal markers such as vimentin and alpha-smooth muscle actin (α-SMA) culminates EMT to a morphologically stable mesenchymal phenotype. At the molecular level, the EMT regulatory machinery involves complex interactions between a core set of EMT-activating transcription factors (EMT-TFs), which include SNAI1; SNAI2; TWIST1; ZEB1 and ZEB2; and EMT-suppressive factors (GRHL2, OVOL1/2, miR-200 family, etc.) [14,15,16,17].

Although the activation of EMT-TFs ultimately results in cellular dedifferentiation (mesenchymal phenotype), each of the EMT-TFs is known to operate in a specific and nonredundant manner [18]. In particular, SNAI1 is a member of the Snail superfamily of transcription factors, implicated in crucial processes such as cellular differentiation, cell movements, and survival [19]. During embryonic development, SNAI1 induces EMT during the formation of the mesoderm and the neural crest, and the functional deletion of SNAI1 is lethal due to gastrulation defects [20]. During cancer-associated EMT, SNAI1 enforces EMT by functioning as a strong repressor of E-cadherin and tight junction components, claudins, while also upregulating proteins of the mesenchymal phenotype, such as vimentin and fibronectin. However, there is a lack of studies focusing on SNAI1-driven molecular targets that are regulated during this early E to IE transition and how these targets are controlled over the course of a complete EMT. Molecular investigations along this line will help us to deduce an operational gene regulatory framework underlying the EMT spectrum.

In the context of EMT, recent research findings have identified novel mechanisms, outlining that specific alterations in the transcriptional signature correlate with changes in chromatin conformational reorganization. For instance, lamin B1 associates with actively transcribing C/G-rich gene regions studded with active histone marks to form clusters known as euchromatin lamin B1-associated domains (eLADs), which change dynamically during transforming growth factor-beta (TGF-β)-induced EMT [21]. In epithelial breast cancer cells, depletion of a subunit of the chromatin-binding cohesin complex, RAD21, causes the reduced transcriptional activity of *TGFB1* and integrin alpha 5 (*ITGA5*) [22]. Furthermore, during the mesenchymal-to-epithelial transition (MET) in mouse mammary gland cells, transcription factors Grhl3 and Hnf4α induced the formation of DNA-looping structures between the promoter and enhancer sites, which are crucial for activating E-cadherin expression [23]. Although these studies shed light on how changes in chromatin conformation can alter the EMT, there is no data showing how this regulation operates along the EMT spectrum.

By using two ovarian cancer cell lines expressing SNAI1, we demonstrate that SNAI1 represses other major EMT-TFs in epithelial (E) cell lines, whereas this regulatory pattern becomes more relaxed in an intermediate epithelial (IE) cell line. Through a candidate gene approach, we identified *PERP* and *ERBB3* as SNAI1-driven early EMT target genes. Subsequently, we show that the loss of *PERP* and *ERBB3* expression along the EMT spectrum is associated with a sequential reduction in the enrichment of RAD21 binding to their regulatory elements. Using a tetracycline-controlled transcriptional activation (Tet-On) system, we further show that the association of RAD21 is likely to be dependent on the expression of the epithelial gatekeeper, Grainyhead-like 2 (GRHL2).

## 2. Results

### 2.1. Expression of SNAI1 Uncovers the Presence of Sequential EMT Changes Along the EMT Spectrum

To understand the early changes induced by SNAI1, we stably overexpressed full-length *SNAI1* in two ovarian cancer cell lines: one belonging to the epithelial (E) phenotype (OVCA420) and the other belonging to the intermediate epithelial (IE) phenotype (OVCA429). Gene expression from real time-quantitative PCR (RT-qPCR) analyses revealed that *SNAI1* overexpression downregulated the expression of major EMT-TFs such as *SNAI2*, *TWIST1,* and *ZEB1/2* in OVCA420 (E) cells (Figure 1A). However, in OVCA429 (IE), *SNAI1* overexpression showed significant upregulation of *TWIST1* and *ZEB1/2* (Figure 1B), while still suppressing *SNAI2* expression. With respect to *SNAI2* repression in OVCA429, we have shown that SNAI1 predominantly repressed *SNAI2* expression through the recruitment of the histone deacetylation machinery [24]. For other EMT-TFs, the pattern of SNAI1-mediated transcriptional regulation switched from a repressive regulation in E cells to a positive regulation in IE cells (Figure 1C).

At the phenotypic levels, *SNAI1* overexpression in OVCA420 cells did not induce any change in cellular morphology (phase-contrast images) or difference in E-cadherin and F-actin localization (immunofluorescence staining) (Figure 1D, left panel), when compared to its control. However, *SNAI1*-OVCA429 cells showed significant morphological alterations evidenced by the dispersed fibroblastic mesenchymal-like phenotype together with the reorganization of F-actin and the complete loss of E-cadherin expression (Figure 1D, right panel). The observed variation in the cellular morphology was further demonstrated by measuring the internuclear distance of two neighboring cells [25], which showed a modest 1.2×-fold increase in *SNAI1*-OVCA420, and a higher 2×-fold increase in *SNAI1*-OVCA429 when compared to its respective control cells (Figure 1E). The increase in internuclear distance suggests a less compact organization after overexpressing SNAI1 in OVCA420 cells. At the functional level, *SNAI1*-OVCA420 showed a lower migration potential (Figure 1G), while *SNAI1*-OVCA429 cells migrated significantly faster (Figure 1H) when compared to their respective controls. Moreover, *SNAI1*-OVCA420 cells showed increased proliferation and lower resistance to anoikis, whereas *SNAI1*-OVCA429 cells showed decreased proliferation, decreased invasion, and higher resistance to anoikis when compared to their respective vector controls (Appendix A). Taken together, SNAI1 overexpression in E cells showed a repressive effect on other EMT-TFs, which resulted only in a subtle increase of internuclear distances. However, in the IE phenotype, the SNAI1-mediated activation of EMT-TFs resulted in a series of morphological changes such as the increase of internuclear distances, enhanced migratory potential, and a transition towards a fibroblastic-like phenotype. These results clearly indicated that SNAI1 overexpression rendered differential EMT-associated phenotypic changes depending on the epithelial status of the cell lines.

### 2.2. ERBB3 and PERP as Targets of SNAI1 Induced Sequential EMT Changes

In order to further characterize the molecular regulation underlying the SNAI1-induced differential EMT-associated phenotypic changes, we chose the gradual increase in internuclear distance (Figure 1E) as a phenotypic parameter to construct a SNAI1-driven EMT spectrum model (Figure 2A). Accordingly, EV-OVCA420 cells showed the E phenotype, while *SNAI1*-OVCA420 cells transited to the IE phenotype with the significant increase in the internuclear distance. While EV-OVCA429 cells also demonstrated the IE phenotype, *SNAI1-OVCA429* cells showed classic features of a complete mesenchymal-like (M) phenotype, such as the spindle-shaped morphology, loss of membranous E-cadherin adhesions, rearranged F-actin structures, and increased migratory potential. To delineate the underlying SNAI1-mediated differential transcriptional regulation along this spectrum, we adopted a candidate approach and identified *PERP* and *ERBB3* as potential gene targets (unpublished data). *PERP*, a p53/p63 direct target gene, is a structural component of desmosomes in the stratified epithelium [26]. *ERBB3*, a member of the epidermal growth factor receptor family, is a direct target of the EMT suppressor GRHL2 [27,28] and is lost in the intermediate mesenchymal (IM) and M phenotypes [8].

Our results show that the expression of ERBB3 was significantly downregulated in *SNAI1*-OVCA420 cells at the mRNA and protein levels. However, the expression of PERP was not significantly affected in *SNAI1*-OVCA420 cells (Figure 3B,D). In *SNAI1*-OVCA429, the expression of *PERP* and *ERBB3* was significantly downregulated at both mRNA and protein levels when compared to the EV-OVCA429 levels (Figure 3C,D). Moreover, when normalized to EV-OVCA420, it was apparent that the expression levels of *PERP* and *ERBB3* showed a dose-dependent anticorrelation with the internuclear distances (Figure 3E,F). These results indicate that the transcription of *PERP* and *ERBB3* is altered and started from the early phases of EMT.

### 2.3. Selective Enrichment of Cohesin-Complex Component RAD21 and Transcription Factor GRHL2 on PERP and ERBB3 Loci Modulate Target Gene Transcription along the EMT Spectrum

Our candidate approach showed that both *PERP* and *ERBB3* gene regulatory regions are enriched with DNA-binding protein RAD21 and transcription factor GRHL2. RAD21, a key structural component of the multiprotein, ring-shaped cohesin complex, which plays a crucial role in sister chromatid separation during mitosis and DNA repair [29,30]. Recent findings show that higher order cohesin-mediated chromatin alteration possibly regulates the spatial and temporal regulations of gene transcription [31,32]. GRHL2, a transcription factor that is essential for the maintenance of epithelial phenotype, operates by upregulating epithelial markers, including E-cadherin and microRNA-200 family members, and repressing EMT inducer ZEB1 [27,33]. Moreover, recent findings indicate that GRHL2 regulates the chromatin accessibility of epithelial gene regulatory elements [34,35]. Since RAD21 and GRHL2 have implications in regulating chromatin and gene transcription, we investigated whether transcriptions of *PERP* and *ERBB3* are controlled by RAD21 and GRHL2. Therefore, we assayed for the enrichment of RAD21 and GRHL2 on conserved binding sites on *PERP* (in chromosome 6q23.3) and *ERBB3* (in chromosome 12q13.2) loci. Accordingly for the *PERP* loci, we designed three amplicons in the promoter/5′UTR (P4, 5, and 7); one amplicon in the intron1 (P12); and three amplicons in the 3′UTR/poly-A-tail regions (P18, 19, and 20) (Figure 3A and Appendix A). Through chromatin immunoprecipitation (ChIP), we identified a significant enrichment of RAD21 and GRHL2 among all three regions (promoter/5′UTR, intron1, and 3’UTR/poly-A-tail) in E cells (EV-OVCA420; Figure 3C,D). Interestingly, in IE cells (*SNAI1*-OVCA420 and EV-OVCA429), enrichment of RAD21 and GRHL2 was higher in two regions (promoter/5′UTR and 3’UTR/poly-A-tail), whereas the enrichment was much lower in the intron1 region (Figure 3C,D). Furthermore, the mesenchymal-like cell line (*SNAI1*-OVCA429) lacking GRHL2 expression (Figure 3D) showed a drastic loss of RAD21 enrichment in the intron1 and 3′UTR/poly-A-tail regions, while it still preserved the RAD21 enrichment in the promoter/5′UTR along the *PERP* loci (Figure 3C).

Similarly, for the *ERBB3* loci, we designed four amplicons in the promoter/5′UTR (E4 and E7–9) and two amplicons in the intron1 region (E19 and 20) (Figure 3B). In the E (EV-OVCA420) and IE (*SNAI1*-OVCA420 and EV-OVCA429) cells, the enrichment of RAD21 and GRHL2 was significantly higher in promoter/5′UTR and intron1 regions (Figure 3E,F). However, similar to the *PERP* loci, the mesenchymal-like cell line (*SNAI1*-OVCA429) failed to show RAD21 enrichment in the intron1 region, with only a slight enrichment in the promoter/5′UTR regions. In summary, these results indicated that the enrichment of RAD21 on the early EMT response genes significantly varied along the EMT spectrum. The specific loss of RAD21-binding at the distal enhancer sites occurred early during the phase transitions, which resulted in reduced transcriptional and, subsequently, translational activity, leading to a mesenchymal-like phenotype.

Using an ovarian cancer four-cell-line model, we have previously established that PEO1 (E), OVCA429 (IE), SKOV3 (IM), and HEYA8 (M) could represent the different EMT states along the spectrum [27] (Appendix A). Furthermore, our in-house ChIP-seq data from five major histone H3 marks (H3K4me3, H3K4me1, H3K9me3, H3K27me3, and H3K27ac) of these cell lines have led us to generate a robust ChromHMM analysis to predict the chromatin accessibility across different gene regulatory regions, including promoters and enhancers [35] (Appendix A). Therefore, we explored the ChromHMM states on *PERP* and *ERBB3* loci in this four-cell-line EMT model to validate the results obtained in the *SNAI1*-mediated overexpression systems. Accordingly, our ChromHMM results showed that chromatin states remained active states at the promoter/5′UTR (P4, 5, and 7) and the intron1 (P12) amplicon regions of the *PERP* loci, across the cell lines of the EMT spectrum (Appendix A). However, at the 3′UTR/poly-A-tail (P18, 19, and 20) amplicon regions of the *PERP* loci, the chromatin landscape switched from an active state in PEO1 and OVCA429 to a partially active state in SKOV3 and to a primed state in HEYA8 (Appendix A). In the case of a *ERBB3* loci analysis, chromatin states remained active in the promoter/5′UTR (E4 and E7–9) for almost all cell lines, with a partial poised/bivalent state in HEYA8 (Appendix A). Subsequently, chromatin states of the intron1 region (E19 and 20) drastically shifted from an active state in PEO1 and OVCA429 to a repressed state in SKOV3 and HEYA8 (Appendix A). Therefore, by using this four-cell-line EMT spectrum model, we could show that the differential enrichment of histone modifications might alter chromatin accessibility for the recruitment of transcription machinery to ultimately regulate gene expression levels.

### 2.4. Enrichment of RAD21 on PERP and ERBB3 Loci Is Partially Dependent on the Abundance of GRHL2 Expression

In addition to the selective loss of RAD21 enrichment at the distal enhancer sites, we speculated whether levels of RAD21 enrichment could be partially dependent on GRHL2 recruitment to the same region. Given the importance of GRHL proteins in controlling chromatin conformation structures during the cellular transition [23,34] and reinforcing epithelial differentiation [27,33,35,37], we tested whether GRHL2 would play a role in altering the RAD21-mediated enrichment along the *PERP* and *ERBB3* loci in our EMT spectrum model. To verify this, we used a shRNA-mediated GRHL2 knockdown in OVCA429 cells, which showed significantly reduced expressions of *ERBB3* at the mRNA and protein levels, with a modest reduction of *PERP* levels (Figure 4A).

Accordingly, upon doxycycline treatment, tetracycline-controlled GRHL2 rescue (mutant GRHL2* expression that is resistant to shGRHL2 [35]) in OVCA429 cells showed a significant increase in *ERBB3* and a modest increase *PERP* transcript levels (Figure 4A). Subsequently, we assessed the enrichment of RAD21 and GRHL2 along the above-mentioned *PERP* and *ERBB3* loci in the inducible system. At the *PERP* loci, we identified a significant loss of RAD21 enrichment in the amplicons encompassing the promoter/5′UTR (P4) and the intron1 (P12) regions in GRHL2 knockdown cells (Figure 4B, left). In the same samples, a similar loss of RAD21 enrichment was observed for half of the promoter/5′UTR (E4 and E7) and the intron1 (E20) amplicon regions along the *ERBB3* loci (Figure 4B, right). Upon GRHL2 rescue, we observed a significant increase in RAD21 enrichment for the promoter/5′UTR (P5 and 7) and the 3′UTR/poly-A-tail (P18) amplicon regions along the *PERP* loci (Figure 4C, left), and all of the promoter/5′UTR (E4 and E7–9) and the intron1 (E19 and 20) amplicon regions along the *ERBB3* loci (Figure 4C, right). Taken together, these results highlighted that GRHL2 plays an important role in determining RAD21-mediated chromatin regulation in epithelial cells, whereas the loss of GRHL2 expression potentially affects the binding of cohesin complexes such as RAD21 to the chromatin.

## 3. Discussion

As established by several decisive studies, the activation of EMT-TFs and their downstream targets in cancer cells primarily result in the detachment of cellular adhesions and the loss of cellular polarity, culminating towards the acquisition of a mesenchymal-like cellular architecture [17,38,39]. Using two SNAI1 overexpression clones, we have shown that the SNAI1-mediated regulation on major EMT-TFs displayed a radical shift from a repressive signal in the E phenotype (OVCA420) towards an activation signal in the IE phenotype (OVCA429). This diversification in SNAI1-mediated regulation denotes two major points: (i) SNAI1 functions as a master inducer of EMT by directly controlling the expression of other EMT-TFs. Several reports have supported this observation. For example, SNAI1 expression in epithelial cells induced the expression of ZEB1 and ZEB2 [40,41]. In a TGF-β model, SNAI1 expression is necessary for ZEB1 expression and TWIST1 protein stabilization [42]. (ii) Depending on the EMT status of the cell lines, SNAI1 operates differentially in controlling the EMT process, including the expression of other EMT-TFs. In our study, overexpression of SNAI1 in a strictly epithelial cell line (OVCA420) generated partial EMT-like changes (such as a subtle increase in the internuclear distance). In contrast, in an intermediate-epithelial cell line (OVCA429), forced SNAI1 expression accelerated the EMT process, as evidenced through changes in the morphology and migratory potential. These results suggest that SNAI1 expression is crucial during the onset of EMT, possibly during the E-to-IE transition. The SNAI1-mediated activation of other EMT-TFs in later transition events further reinforces a mesenchymal phenotype. This is consistent with the data derived from human cancers. When compared with the normal endometrium, *SNAI1* was upregulated as early as in the noninvasive (stage 1A) and myoinvasive (stage 1B) tumors, followed by the expression of *TWIST1*, *ZEB1*, and *SNAI2* in the myoinvasive (stage 1C) tumors of endometrioid endometrial carcinoma [43]. Moreover, our previous report on a panel of 43 ovarian cancer cell lines also showed that SNAI1 expression is enriched in cell lines with E and IE phenotypes, whereas expressions of TWIST1 and ZEB1/2 are higher in cell-line IM and M phenotypes.

The cohesin complex plays a crucial role in facilitating long-range interactions between gene regulatory elements, thereby executing a higher-order spatiotemporal regulation of transcriptions, as evidenced in several vital cellular processes such as DNA repair and proper chromosome segregation during cell division [30]. RAD21, a key clamping component of the ring-like cohesin complex, is correlated with breast and colon cancer progression and is reported to alter the transcriptional activity of EMT-induced mesenchymal genes [22,44,45]. Using our EMT spectrum model, we have shown a sequential loss of RAD21 enrichment along specific gene regulatory regions (promoter and enhancer) of the *PERP* and *ERBB3* loci. Such selective enrichment of RAD21 and GRHL2 along these regulatory elements led us to derive a hypothetical model where we correlate the enrichment of RAD21 and GRHL2 to the formation of chromatin loops in an epithelial-like cell state, and the sequential loss of such enrichments was correlated to a release of preformed chromatin loops in a mesenchymal-like cell state (Figure 5).

According to this hypothetical model, in regards to the *PERP* loci, our RAD21-ChIP analyses could be interpreted as the formation of a proximal loop (~0.7kb) between the promoter/5′UTR (P4, 5, and 7) and the intron1 (P12) with a distal loop (~19kb) between the intron1 (P12) and the 3′UTR/poly-A-tail (P18–20) regulatory regions, exclusively in an epithelial-like cell (EV-OVCA420; Figure 5). These looping structures might have rendered the assembly of regulatory regions in close proximity for transcription, leading to high mRNA/protein levels. Interestingly, in intermediate epithelial-like cells (*SNAI1*-OVCA420 and EV-OVCA429), the preferential loss of RAD21 enrichment at the intron1 (P12) region potentially discharged the proximal loop but still would have maintained the distal looping intact between the promoter/5′UTR (P4, 5, and 7) and the 3′UTR/poly-A-tail (P18–20) regulatory regions. The potential loss of the proximal loop in intermediate epithelial-like cells might explain the reduced transcriptional/translational output when compared to the epithelial-like cell lines (EV-OVCA420). Finally, in the mesenchymal-like cells (*SNAI1*-OVCA429), a total lack of RAD21 binding at the intron1 (P12) and the 3′UTR/poly-A-tail (P18–20) regions might indicate a complete loss of the proximal as well as distal loops, which potentially affected the *PERP* mRNA and protein levels.

In the case of the *ERBB3* loci, the possible formation of a short DNA loop (~2kb) between the promoter/5′UTR (E4 and E7-9) and the intron1 (E19 and 20) regulatory regions might remain unaltered for the epithelial-like (EV-OVCA420) and the intermediate epithelial-like cells (*SNAI1*-OVCA420 and EV-OVCA429). However, in the mesenchymal cells (*SNAI1*-OVCA429), selective loss of RAD21 binding at the intron1 (E19 and 20) region might have resulted in the loss of a DNA loop between the two regulatory regions, which potentially affected the *ERBB3* mRNA/protein levels. These observations are in-line with previous publications showing that long-range interactions between distant cis-regulatory elements are mediated through DNA looping structures, and alterations occurring in these structures implicate a transcriptional output [23,46,47]. Additionally, ChromHMM histone-state analyses revealed that the chromatin state of the promoter-associated regions remained unaltered and open throughout the EMT spectrum, for both the *PERP* and *ERBB3* loci. However, at the distal enhancer regions, a dramatic shift occurred from an active chromatin state in epithelial and intermediate epithelial-like cell lines (PEO1 and OVCA429) to a partially active (in the *PERP*-intron1 region) in the intermediate mesenchymal cell line (SKOV3). This further pushed to a primed (in the *PERP*-3′UTR/poly-A-tail region) or a repressed (in the *ERBB3*-intron1 region) state in the mesenchymal phenotype (HEYA8). Such different chromatin landscapes in multiple EMT intermediate subpopulations have been reported by using an assay for transposase-accessible chromatin-sequencing (ATAC-seq) [7]. Therefore, these differential histone modifications in selective gene regulatory regions further illustrate the dynamic interplay between epigenetic modifications and the transcription machinery along the EMT spectrum.

Another important observation in our RAD21-mediated chromatin regulation is the partial dependency of transcription factor GRHL2 in the assayed gene regulatory regions. GRHL2 and other Grainyhead family members act as prime determinants of the epigenetic alterations of epithelial genes and function as pioneer factors to establish cell type-specific accessible chromatin landscapes, such as the opening of epithelial enhancers [23,34,35,48]. In accordance with these reports, our results showed that GRHL2-lacking mesenchymal cell lines failed to retain RAD21-mediated promoter-enhancer interactions, which remained intact in the epithelial-like and intermediate epithelial-like cell lines. Moreover, by using our GRHL2 knockdown and inducible overexpression systems, we were able to show the partial rescue of RAD21 recruitment back to those regulatory regions potentially to modulate candidate gene expressions. Based on the current findings and previous studies, we propose a hypothetical model (Figure 5) that GRHL2/cohesin-mediated DNA looping would be required for transcriptional accessibility and epigenetic modifications at the epithelial gene loci. Such a novel regulation seems to be crucial for maintaining and reinforcing the epithelial trait. This proposed hypothetical model would point to the future research direction focusing on genome-wide long-range chromatin looping changes during EMT.

## 4. Materials and Methods

### 4.1. Cell Culture and Generation of Stable Cell Lines

Ovarian cancer cell lines OVCA420 and OVCA429 were cultured in DMEM containing 10% FBS. For stable overexpression of *SNAI1*, full-length wild-type *SNAI1* (#LVP314779) was purchased from Origene (Rockville, MD, USA) and cloned into pLenti-GIII-CMV-GFP-2A-Puro backbone (#LV590, all from Applied Biological Materials Inc (ABM), Richmond Canada). Empty vector with no inserts was used as a negative control. Plasmids were mixed with MISSION Lentiviral Packaging Mix (#SHP001; Sigma-Aldrich, St. Louis, MO, USA) and transfection reagent Fugene 6 (#11814443001; Roche, Mannheim, Germany) prior to adding 293T cells. Forty-eight and 72 h post-transfection, virus-containing supernatants were harvested, filtered, and added to OVCA420 and OVCA429, along with 8-μg/mL polybrene (Sigma Aldrich). Twenty-four hours after infection, cells were subjected to puromycin selection at a concentration of 6 to 7 μg/mL. Stable cell lines were maintained with 5 μg/mL of puromycin. OVCA429 cells carrying GRHL2 targeting stable short hairpin RNA (shGRHL2 #12), control-plasmid luciferase shRNA (shLuc), and the Tet-On GRHL2 overexpression plasmid containing mutated GRHL2* (resistant to shGRHL2 #12) were utilized from our previous studies [27,35]. GRHL2 expression was induced through 1-μg/mL doxycycline for 96 h.

### 4.2. Real Time-Quantitative PCR (RT-qPCR) and Chromatin Immunoprecipitation-qPCR (ChIP-qPCR)

Total RNA was extracted using the RNeasy mini kit (SAbiosciences, Qiagen, Toronto, Canada). For gene expression RT-qPCR, 1-μg RNA was reverse-transcribed to cDNA using RT^2^ First Strand kit (SAbiosciences, Qiagen) and mixed with SYBR green master mix (SAbiosciences, Qiagen) for qPCR. Three out of five housekeeping gene expressions (*ACTB*, *B2M*, *GAPDH*, *HPRT1*, and *RPL13A*) were used for normalization. mRNA expression level from a minimum of three biological replicates was normalized to the average of housekeeping genes’ expressions and presented as relative mRNA levels (2^−∆Ct^) or as an average fold change (2^−∆∆Ct^) relative to its respective vector control.

For ChIP assays, about 30–50 million cells were cross-linked with 1% formaldehyde for 10 min and quenched by adding 0.125-M glycine for 5 min. Cells were washed twice and harvested with 1× PBS and subsequently lysed in nuclear lysis buffer (50-mM Tris-Cl, 10-mM EDTA, 1% SDS, and protease inhibitors) for 1 h at 4 °C. After lysis, the nuclear pellet was sonicated on ice to achieve chromatin sizes of 200 bp to 500 bp suspended in IP buffer, which contains ChIP dilution buffer (50-mM Tris-Cl, 0.167-M NaCl, 1% Triton X-100, 0.02% NaN_3_, and protease inhibitors) and nuclear lysis buffer in a 9:1 ratio. About 25 µg of sheared chromatin per immunoprecipitation was subsequently incubated with 2 µg of primary antibody overnight at 4 °C. Post-incubation, samples were mixed with protein G sepharose beads (GE Healthcare, Chicago, IL, USA) and incubated for an additional 4–6 h at 4 °C. Crosslinked protein-DNA complexes were reverse-crosslinked in 1% SDS/0.1-M NaHCO_3_ by heating at 65 °C. The resulting DNA was eluted using the QIAquick PCR purification kit (Qiagen) following the manufacturer’s protocol. Purified samples and 1% input controls were used for qPCR analysis. Primers and antibodies used for ChIP and ChIP-qPCR, respectively, were provided in Appendix A, respectively.

### 4.3. Western Blotting

Whole-cell lysates were prepared using RIPA buffer and resolved by standard reducing SDS-PAGE followed by blotting on polyvinylidene difluoride (PVDF) membranes. Immunoblots were incubated overnight with suitable primary antibodies diluted in 2% BSA in Tris-buffered saline with 0.1% Tween (TBST). Infrared dye-conjugated secondary antibodies from Li-COR Biosciences, Lincoln, NE, USA at 1:10,000 diluted in TBST: IRDye 800-CW goat anti-mouse or anti-rabbit (#926-32210 and #926-32211) and IRDye 680LT goat anti-mouse or anti-rabbit (#926-68020 and #926-68021). Blots were scanned using the Odyssey infrared imaging system (Li-COR), and the resulting images were transferred to greyscale. Antibodies used for Western blotting were provided in Appendix A.

### 4.4. Immunofluorescence Staining and Analysis of Internuclear Distance

Cells were seeded on 15-mm glass coverslips to appropriate confluence, fixed with 4% paraformaldehyde for 10 min, and quenched with 0.05% Triton-X for 5 min for membrane permeabilization. The fixed cells were incubated with blocking buffer (3% BSA in PBS) for 1 h and stained with E-cadherin primary antibody overnight at 4 °C. Alexa Fluor 488-conjugated anti-mouse (#A11029) and Alexa Fluor 594-conjugated anti-mouse (#A11032) secondary antibodies from Invitrogen, California, CA, USA were used for visualization. For F-actin staining, rhodamine-conjugated phalloidin (#R415; Life Technologies, California, CA, USA) was used directly after blocking. The stained coverslips were mounted onto glass slides using Vectashield mounting medium containing DAPI (#H-1200) from Vector Laboratories, Burlingame, CA, USA.

To quantify the internuclear distance, random images of each EMT-TFs overexpression cells and its respective control cells were imaged at 60× magnification using a Nikon A1R confocal microscope. The distance between two adjacent nuclei (in µm) was quantified using NIS Software (Nikon Instrument Inc, Melville, NY, USA). A minimum of 100 nuclear distances was taken to obtain the mean internuclear distance for each condition.

### 4.5. Gap Closure Migration Assay

For the gap closure migration assay, culture inserts (#80209; Ibidi, Martinsried, Germany) were used to make the gaps in a 6-well culture plate. Cells were seeded into both chambers of the culture inserts at 100% confluent. After removal of the culture insert, cells were washed with PBS and supplemented with fresh culture medium. Live cell tracking was documented by using the Nikon C1 live-imaging system.

## 5. Conclusions

In summary, SNAI1-mediated core EMT-TFs’ regulation switches from a repressive regulation in cells with an epithelial phenotype to an activation pattern in cells with an intermediate epithelial phenotype. Detailed analysis of two candidate genes, PERP and ERBB3, showed that selective enrichment of RAD21 and GRHL2 on the gene regulatory elements modulate their expression levels. Results from this study suggest a proposed model where early EMT genes are subjected to differential chromatin looping alterations, and these structures are partially dependent on the expression of epithelial gatekeeper GRHL2.

## Figures and Tables

**Figure 1 cancers-12-01140-f001:**
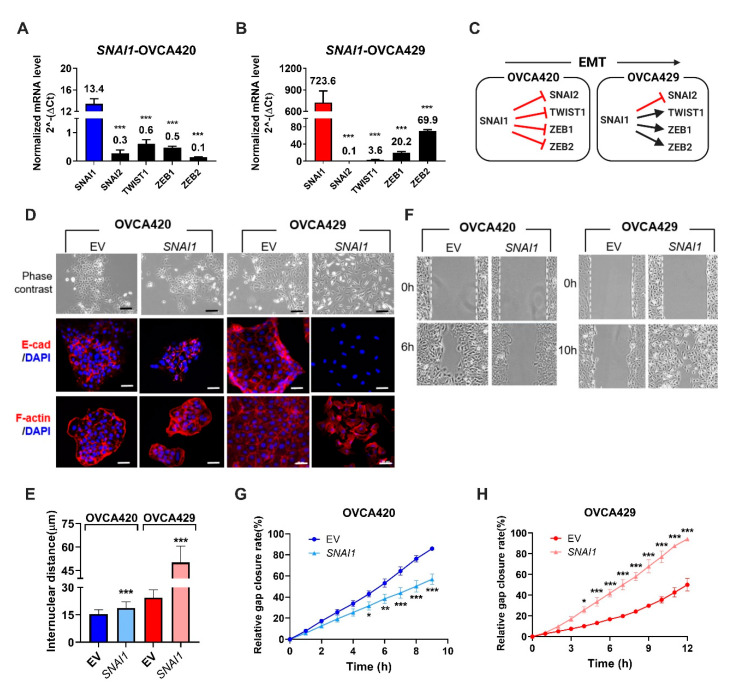
*SNAI1* overexpression in ovarian cancer cells induces differential morphological and phenotypic changes. (**A**,**B**) Bar charts showing normalized mRNA levels of *SNAI1*, *SNAI2*, *TWIST1*, *ZEB1*, and *ZEB2* mRNA expression (*y*-axis) in an epithelial cell line; OVCA420 (**A**); and an intermediate epithelial cell line OVCA429 (**B**) stably overexpressing *SNAI1*. Mean ± SEM from three independent experiments. One-way ANOVA and Bonferroni post-hoc tests were used; *** *p* < 0.001 (**C**) Diagram illustrating a SNAI1-induced transcriptional regulatory network model using OVCA420 and OVCA429. The mRNA levels (from Figure 1A,B) ≤2 are considered as downregulated (red blocks), while levels ≥2 are considered as upregulated (black arrows). Illustration created with Biorender.com. (**D**) Phase-contrast images (top row), immunofluorescence staining of E-cadherin (red color, middle row), and F-actin staining (red color, bottom row) together with DAPI (blue color) of control (EV) and SNAI1-overexpression clones. Scale bars indicate 200 μm for phase-contrast images and 50 μm for immunofluorescence images. (**E**) Bar charts showing mean internuclear distance (*y*-axis, μm) of control (EV) and SNAI1-overexpression clones. Error bars represent SEM. Unpaired *t*-tests were used; *** *p* < 0.001. (**F**) Phase-contrast images from wound-healing migration assays showing gap closure of control (EV) and SNAI1-overexpression clones at indicated timepoints. White dotted lines denote the edges of the initial gaps. (**G**,**H**) Line graphs of control (EV) and *SNAI1*-overexpression clones, showing the percentage of gap area covered by cells (y-axis) at different time intervals (x-axis). Mean ± SD from two independent experiments. Two-way ANOVA and Bonferroni post-hoc tests were used. * *p* < 0.05, ** *p* < 0.01, and *** *p* < 0.001. EMT: epithelial-mesenchymal transition. EV: control.

**Figure 2 cancers-12-01140-f002:**
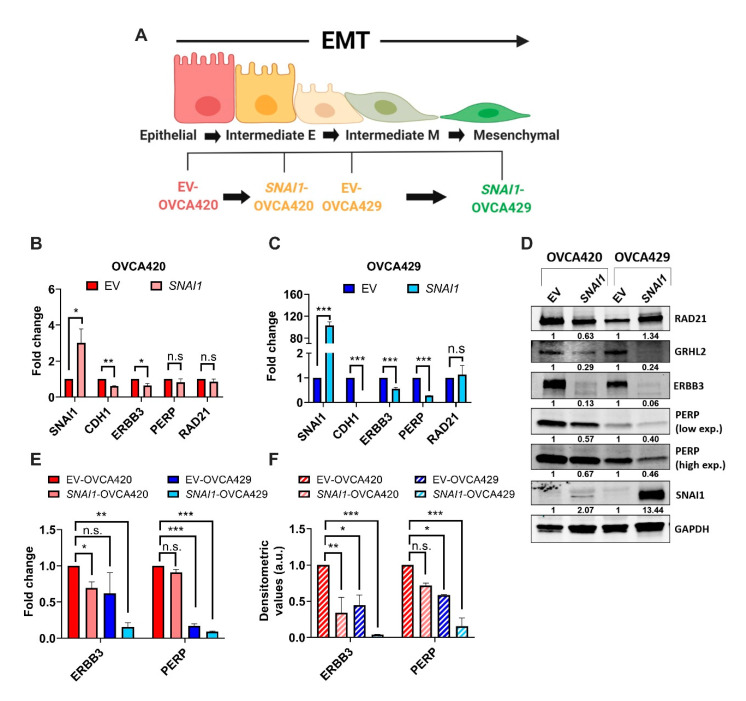
Downregulation of early EMT targets in SNAI1-overexpression cancer cell lines. (**A**) Illustration showing the EMT spectrum model: SNAI1-overexpressing cell lines (SNAI1-OVCA420 and SNAI1-OVCA429 and respective controls EV-OVCA420 and EV-OVCA429). Illustration created with Biorender.com. (**B**,**C**) Histograms showing the mean mRNA fold changes of indicated genes in *SNAI1*-expressing OVCA420 (**B**) and OVCA429 (**C**) relative to the respective control (EV) cells, quantified through RT-qPCR. Mean ± SEM from three independent experiments. Unpaired *t*-tests were performed; * *p* < 0.05, ** *p* < 0.01, and *** *p* < 0.001; n.s., not significant. (**D**) Immunoblots showing the expression of indicated proteins in the control (EV), *SNAI1*-OVCA420, and *SNAI1*-OVCA429 cells. Numbers below the gel lanes represent the protein levels, normalized to GAPDH expression and relative to the respective control (EV) cells, determined using ImageJ software. Full-length blots are available in Appendix A. (**E**) Histograms showing the mean mRNA fold changes (from Figure 2B,C) of *ERBB3* and *PERP* of the indicated cell lines, normalized to EV-OVCA420 cells. Mean ± SEM from three independent experiments. Unpaired *t*-tests were performed; * *p* < 0.05, ** *p* < 0.01, and *** *p* < 0.001; n.s., not significant. (**F**) Histograms showing the mean densitometric values of *ERBB3* and *PERP* in indicated cell lines relative to EV-OVCA420 cells, determined using ImageJ software (data from independent duplicates). Mean ± SEM from two independent experiments. Unpaired *t*-tests were performed to compute the statistical significance; * *p* < 0.05, ** *p* < 0.01, and *** *p* < 0.001; n.s., not significant.

**Figure 3 cancers-12-01140-f003:**
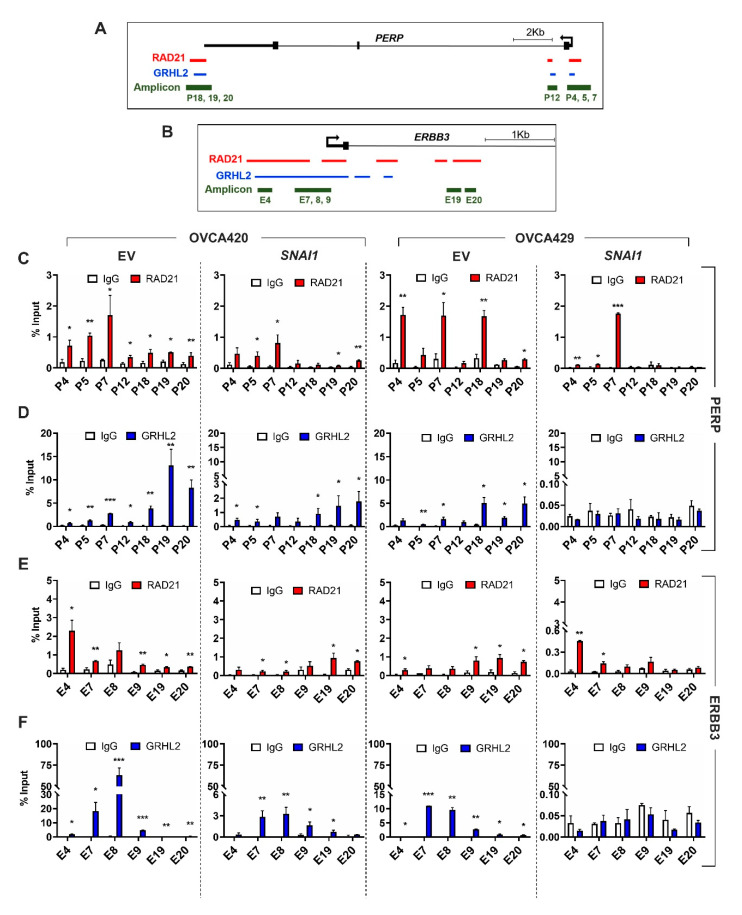
Enrichment of RAD21 and GRHL2 among regulatory regions on *ERBB3* and *PERP* loci significantly varies along the EMT spectrum. (**A**,**B**) Schematic representation of human *PERP* (**A**) on chromosome 6q23.3 and *ERBB3* (**B**) on chromosome 12q13.2, with pointed arrows indicating the transcription start site. RAD21 (red), GRHL2 (blue) binding sites, and the regions amplified (amplicons, green) after chromatin immunoprecipitation (ChIP) were depicted. ChIP-seq-binding regions (obtained from ReMap 2018 [36]). (**C**–**F**) ChIP assay was performed for *SNAI1*-overexpressing OVCA420 and OVCA429 cells and respective control cells (EV) using antibodies specific for RAD21 (red, **C**,**E**); GRHL2 (blue, **D**,**F**); or IgG (white, **C**–**F**) on *PERP* (**C**,**D**) or *ERBB3* (**E**,**F**) gene loci. Signals were normalized to input DNA (1%) and plotted as enrichments relative to their respective IgG control. Results for at least two biological replicates were shown ± SEM. Unpaired *t*-tests were performed to compute statistical significances; * *p* < 0.05, ** *p* < 0.01, and *** *p* < 0.001.

**Figure 4 cancers-12-01140-f004:**
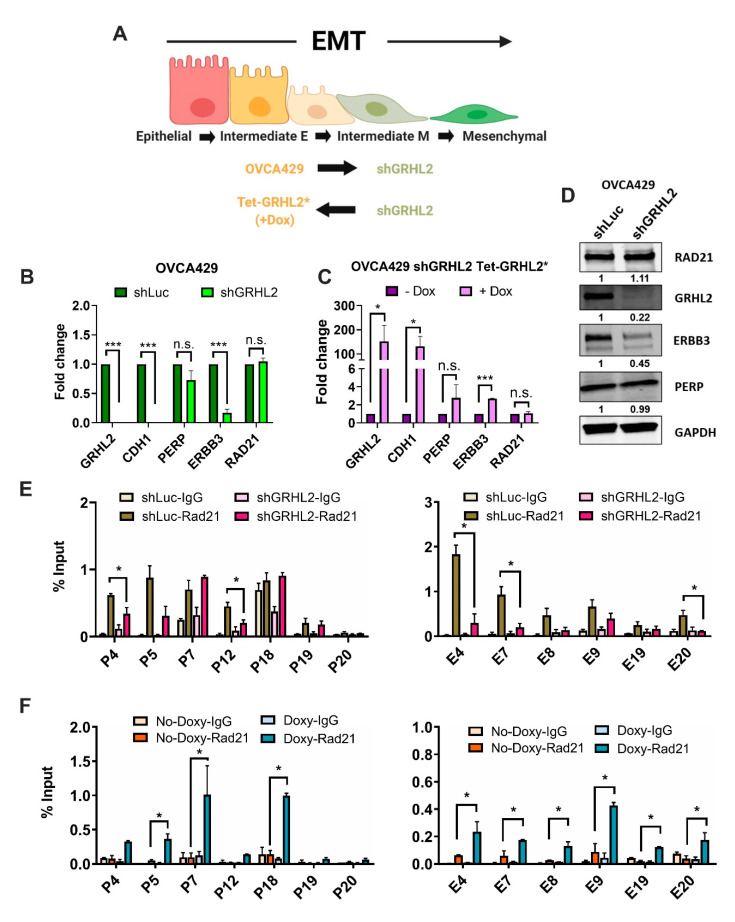
GRHL2 levels partially determine RAD21 recruitment on *PERP* and *ERBB3* loci. (**A**) Illustration showing the EMT spectrum model: OVCA429 GRHL2-knockdown (shGRHL2) and OVCA429 shGRHL2 Tet-GRHL2* (rescue) cells. Illustration created with Biorender.com. (**B**,**C**) Expression levels of indicated genes in control (shLuc) and GRHL2-knockdown (shGRHL2) OVCA429 cells (**B**) and OVCA429 shGRHL2 Tet-GRHL2* (**C**) cells with/without 96-h doxycycline-induced GRHL2 overexpression as analyzed through RT-qPCR. Results for at least three biological replicates were shown ± SEM. Unpaired *t*-tests were performed; * *p* < 0.05, ** *p* < 0.01, and *** *p* < 0.001. (**D**) Western blots of RAD21, GRHL2, ERBB3, and PERP in control (shLuc) and GRHL2-knockdown (shGRHL2) OVCA429 cells. GAPDH was used as a loading control in Western blotting. Full-length blots are available in Appendix A. (**E**,**F**) ChIP-qPCR plots showing RAD21 enrichment levels on *PERP* (**left**) and *ERBB3* (**right**) loci measured in control (shLuc), GRHL2-knockdown (shGRHL2) OVCA429 cells (**E**), and OVCA429 shGRHL2 Tet-GRHL2* (rescue) cells with/without 96-h doxycycline-induced GRHL2 overexpression (**F**). Signals were normalized to input DNA (1%) and plotted as enrichments relative to its respective IgG control. Results for at least two biological replicates were shown ± SEM. Unpaired *t*-tests were performed to compute statistical significance; * *p* < 0.05.

**Figure 5 cancers-12-01140-f005:**
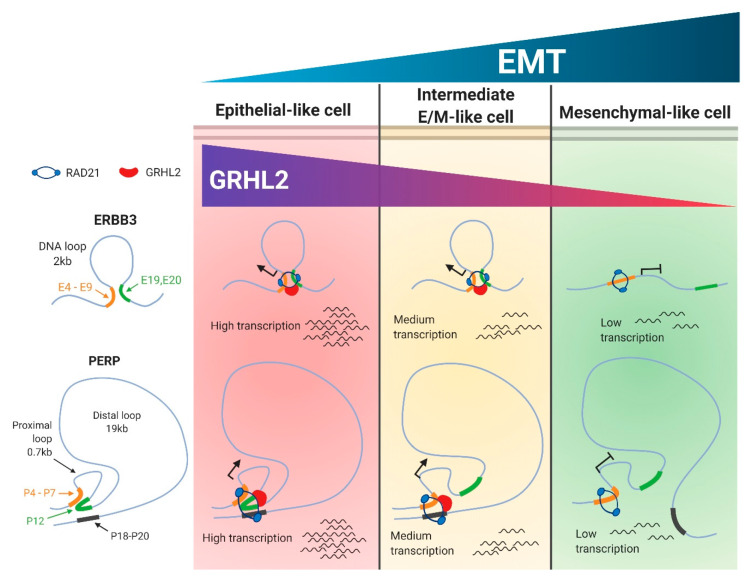
A hypothetical model illustrates that the gradual decrease in *PERP* and *ERBB3* transcript levels along the EMT spectrum potentially correlates with the selective loss of RAD21 and GRHL2-mediated DNA looping structures. In an epithelial cell state, GRHL2 associates with RAD21 possibly to mediate intact chromatin loop structures bringing proximal and distal regulatory elements into the *ERBB3* (**top**) and *PERP* (**bottom**) loci, and these structures might render abundant transcriptions. In the intermediate E/M-like cell state, the selective loss of RAD21/GRHL2 enrichment in the P12 amplicon region of the *PERP* loci potentially leads to the dissociation of proximal loop structures. There is no significant change in the *ERBB3* loci. However, as these cells transit into a mesenchymal cell state, the loss of GRHL2 expression might lead to the distortion of cohesin structures connecting proximal and distal regulatory elements, which may result in the reduced transcriptional activity. Illustration created with Biorender.com.

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
