# Peer review of "SNAI1-Driven Sequential EMT Changes Attributed by Selective Chromatin Enrichment of RAD21 and GRHL2"

_cancers, 2020, doi:10.3390/cancers12051140_

Round 1

Reviewer 1 Report

Authors address the topic of epithelial to mesenchymal transition in cancer cells focusing on the dynamism of the molecular mechanisms driving the progressive acquisition of the mesenchymal phenotype. They take advantage by the use of ovarian cancer cell lines representative of the entire spectrum of phenotypes from the epithelial to mesenchymal one. Their conclusion is that a pivotal role in this dynamic phenomenon is played by GRHL2 that modulates the expression of key epithelial genes such as PERP and ERBB3. The identification of these last genes come from the transcriptional analysis of ovarian cancer cells stably overexpressing single TFs known to be relevant in EMT (Snai1/2, Twist1, Zeb1/2). Finally, in two cancer cell lines overexpressing Snail and representative, with the relative controls, of the entire spectrum of EMT, the authors performed study of chromatin occupancy, concluding that a dynamism in the GRHL-dependent occupancy of the regulatory regions of PERP and ERBB3 by RAD21 redefines transition of cellular states along the EMT spectrum.

Although the work addresses an interesting topic that could reveal mechanisms involved in the acquisition of the metastatic and aggressive phenotype by cancer cells, it presents important critical issues, mainly in the choice   of cellular models.

Major points:

  • The manuscript resulted annoying to read because data and concepts, that would be useful for understanding the text and the experimental path, have been often omitted, or treated too tightly, or referred exclusively to bibliographical references. Some examples: in lane 89 and in fig 1A a SGOCL panel is introduced without any explanation; the same for  CSIOVDB in 110 lane; in lane 210 PERP and ERBB3 have been shown as “prominent candidates” to be transcriptional targets of these EMT-TFs: more information about these factors and some experimental data reserved to the supplementary data need to be inserted in the main body of the text; in lane 287 SKOV3 and HEYA8 appear without any details about them; RAD21, which will take on some importance in the paper, appears for the first time in the lane 260  without any presentation. And so on…
  •  
  • Data presented in Fig 1A have to be better explained: how many cell lines have been analysed and how the statistical analysis has been performed? Moreover, only few cell lines seem to show difference in the transcriptional expression of the TFs, so questioning the biological significance of the data.

  • Data presented in Fig 1 B-D referred to cell lines stably expressing EMT-TFs, used to unveil circuitry of reciprocal transcriptional control. It is difficult to understand why no silencing rather than overexpression experiments have been carried out for this purpose. Furthermore, it is not clear how the data should be interpreted. If the gene expression is presented as fold change, it is assumed that the basal expression value was arbitrary = 1. So, how can a value =1 be statistically significant (i.e. Twist1 in Snai1-PEO1, Snai1 in Zeb2-PEO1, Snai2 in Twist1-OVACA 429)?

  • The most relevant criticism, however, is about the choice of the Snai1-overexpressing cells and the relative controls (OVA420, Snai1OVA420, OVA429, Snai1OVA420) for the study described in fig…... If those cell lines have been chosen because representative of all the phenotypes of the EMT range (E, IE, IM, M) why not use non manipulated cell lines, belonging to the SGOCL panel, for example? In lane 256 the authors affirm “These results indicate that the transcription of PERP and ERBB3 is altered and started from the early phases of EMT”. We think, instead, that the data indicate only that the overexpression of Snai1 is responsible of the morphological changes (Fig. 3A) and of the modification in the expression of CDH1, ERBB3, PERP and RAD21 (Fig. 3B). All data obtained from OVA420, Snai1OVA420, OVA429, Snai1OVA420 has to be confirmed with the use of genetically un-manipulate cells (SGOCL or epithelial cancer cells treated with an EMT-inducing cytokine, as TGFbeta1).

  • The authors explain the result of ChIP analysis on regulative sequences of PERP and ERBB3 with a RAD21-mediated chromatin regulative loop. This conclusion should only be mentioned in discussion. If, however, the authors want to keep it in the abstract and repeatedly in the text, until to propose a model in Fig. 5 they must carry out further experiments of chromosome conformation capture that could prove it.
  •  

In conclusion, the work, while addresses an interesting topic, shows serious criticisms that impede the publication in the present form. Firstly, there are too many approximations and too many data in supplementary information without sufficient explanation in the main text. Secondly, and more critical, the choice of cell lines used is questionable and the statistical analysis unclear.

Author Response

Response to Reviewer 1 comments

1) The manuscript resulted annoying to read because data and concepts, that would be useful for understanding the text and the experimental path, have been often omitted, or treated too tightly, or referred exclusively to bibliographical references. Some examples: in lane 89 and in fig 1A a SGOCL panel is introduced without any explanation; the same for CSIOVDB in 110 lane;

We agree with the reviewer that the flow of the manuscript was not easy to follow. We have removed the data on the SGOCL panel and the CSIOVDB on the revised manuscript.

2) in lane 210 PERP and ERBB3 have been shown as “prominent candidates” to be transcriptional targets of these EMT-TFs: more information about these factors and some experimental data reserved to the supplementary data need to be inserted in the main body of the text

We understand the reviewer’s comments on bringing the necessary supplementary data, but based on the comments from other reviewers, we have decided to remove the first part of the study and reserve it as unpublished data. Therefore, we have indicated PERP and ERBB3 as potential gene targets derived from a candidate approach (lines 152-157 of the revised manuscript).

3) in lane 287 SKOV3 and HEYA8 appear without any details about them;

We agree with the reviewer’s comment. We have explained the significance of these cell lines and also added a supplementary figure explaining the EMT status of these cell lines (lines 231-233, Supplementary figure S2A)

4) RAD21, which will take on some importance in the paper, appears for the first time in the lane 260 without any presentation. And so on…

We agree with the reviewer’s comment. Therefore, we have elaborated RAD21 in section 2.2 with adequate references (Lines 187-191)

5) Data presented in Fig 1A have to be better explained: how many cell lines have been analysed and how the statistical analysis has been performed? Moreover, only few cell lines seem to show difference in the transcriptional expression of the TFs, so questioning the biological significance of the data.

We have generated a new Figure 1 that does not include the SGOCL panel. Regarding the details about the statistical analysis, we have now explained statistical method used for generating every graph in the description below every figure.

6) Data presented in Fig 1 B-D referred to cell lines stably expressing EMT-TFs, used to unveil circuitry of reciprocal transcriptional control. It is difficult to understand why no silencing rather than overexpression experiments have been carried out for this purpose. Furthermore, it is not clear how the data should be interpreted. If the gene expression is presented as fold change, it is assumed that the basal expression value was arbitrary = 1. So, how can a value =1 be statistically significant (i.e. Twist1 in Snai1-PEO1, Snai1 in Zeb2-PEO1, Snai2 in Twist1-OVACA 429)?

We have now generated a new Figure 1A, B by extracting some of the data from Figure 1B-D in the previous version. The gene expression levels plotted in the graph were basal mRNA levels normalized to mean housekeeping gene expression levels. We have mislabelled it as fold change in the previous version, and have now corrected as “Normalized mRNA level – 2^-(ΔCt)”.

7) The most relevant criticism, however, is about the choice of the Snai1-overexpressing cells and the relative controls (OVA420, Snai1OVA420, OVA429, Snai1OVA420) for the study described in fig…... If those cell lines have been chosen because representative of all the phenotypes of the EMT range (E, IE, IM, M) why not use non manipulated cell lines, belonging to the SGOCL panel, for example? In lane 256 the authors affirm “These results indicate that the transcription of PERP and ERBB3 is altered and started from the early phases of EMT”. We think, instead, that the data indicate only that the overexpression of Snai1 is responsible of the morphological changes (Fig. 3A) and of the modification in the expression of CDH1, ERBB3, PERP and RAD21 (Fig. 3B). All data obtained from OVA420, Snai1OVA420, OVA429, Snai1OVA420 has to be confirmed with the use of genetically un-manipulate cells (SGOCL or epithelial cancer cells treated with an EMT-inducing cytokine, as TGFbeta1).

We agree with the reviewer that the observed changes in PERP and ERBB3 transcription levels were SNAI1-mediated effects. Therefore, in the revised manuscript, we have re-written almost the entire manuscript as a SNAI1-derived transcriptional regulation and so on.

8) The authors explain the result of ChIP analysis on regulative sequences of PERP and ERBB3 with a RAD21-mediated chromatin regulative loop. This conclusion should only be mentioned in discussion. If, however, the authors want to keep it in the abstract and repeatedly in the text, until to propose a model in Fig. 5 they must carry out further experiments of chromosome conformation capture that could prove it.

We agree with the reviewer’s comment on validating the RAD21-mediated regulation of PERP and ERBB3 using 3C experiments. However, due to the COVID-19 situation, our research labs are temporarily closed down, limiting our experimental validation. Therefore, we have restricted our proposed model on chromatin looping regulation to the Discussion section, which needs further validation.

Reviewer 2 Report

Sundararajan et al investigated the interplay between EMT-activating transcription factors (EMT-TFs) during the various phases of epithelial to mesenchymal transition of ovarian cells. They used three ovarian cancer cell lines with different epithelial characteristics, to identify cross-regulatory relationships among EMT-TFs through their stable overexpression. Gene expression analysis of the EMT-TF-overexpressing cell lines led the authors to focus on PERP and ERBB3, two prominent target-genes of the EMT-TFs and analyze the mechanism of their repression during EMT. The latter experiments centered on two chromatin regulators, RAD21 and GRHL2 that were shown to mediate long range chromatin interactions implicated in the repression of PERP and ERBB3 during EMT. The authors performed an extensive and technically sound examination of transcription regulatory events associated with ovarian EMT. However, the study lacks focus and fails to clearly formulate a specific scientific question and address it satisfactorily. In the first part of the study the authors analyze the cross-regulation of EMT-TFs and identify differences in their interactions during progression from the epithelial to mesenchymal state. This part is primarily descriptive and does not provide a meaningful conclusion to advance our understanding of the EMT molecular mechanism. Furthermore it is not substantially linked to the second part which focuses on the mechanism of PERP and ERBB3 transcription regulation. The second part could be the basis for a report that could be published and it should focus on the EMT-associated repression of specific genes by RAD21 and GRHL2 dependent chromatin interactions. I have the following specific issues that should be addressed by the authors:

  1. The authors indicate that the expression pattern of EMT-TFs in the cell panel they use does not correlate with their pattern of expression in ovarian tumors (lines 114-117). This is a very problematic fact that could indicate that the tissue complexity in vivo could not be mimicked by the in vitro culture conditions. The authors’ claim that the present study was prompted in order to explain this difference does not make sense and more importantly the results of the study do not help us to understand the difference in EMT-TF expression between the cell lines and tumors.
  2. The analysis of RAD21 and GRHL2 binding to PERP and ERBB3 is substantial and technically sound. In order to substantiate the claims of the authors about the significance of specific chromatin interactions for the repression of PERP and ERBB3 during EMT it would be important to mutate by CRISPR/Cas9-mutagenesis specific binding sites of RAD21 and/or ERBB3 in these genes to compromise critical interactions for looping and demonstrate their functional importance.
  3. Can the authors extend their observations about the role of RAD21 and GRHL2 on PERP and ERBB3 in other cell lines of their collection?

 Minor points

  1. The authors should describe the collection and characteristics of SGOCL in a supplementary file.
  2. In the last paragraph of the results section (lines 334-348) the panels of figure 4 have been mislabeled.
  3. In line 390 something must be missing in the part “i.e. the intermediate-IE cell lines.”

Author Response

Response to Reviewer 2 comments

1) The authors indicate that the expression pattern of EMT-TFs in the cell panel they use does not correlate with their pattern of expression in ovarian tumors (lines 114-117). This is a very problematic fact that could indicate that the tissue complexity in vivo could not be mimicked by the in vitro culture conditions. The authors’ claim that the present study was prompted in order to explain this difference does not make sense and more importantly the results of the study do not help us to understand the difference in EMT-TF expression between the cell lines and tumors.

We agree with the reviewer and in the revised manuscript we have removed the above mentioned data derived from ovarian cancer patients, and any explanation that correlated cell lines with tumor derived data.

2) The analysis of RAD21 and GRHL2 binding to PERP and ERBB3 is substantial and technically sound. In order to substantiate the claims of the authors about the significance of specific chromatin interactions for the repression of PERP and ERBB3 during EMT it would be important to mutate by CRISPR/Cas9-mutagenesis specific binding sites of RAD21 and/or ERBB3 in these genes to compromise critical interactions for looping and demonstrate their functional importance.

We agree with the reviewer’s suggestion on validating RAD21-mediated PERP and ERBB3 regulatory regions using CRISPR/Cas9 mutagenesis method. However, due to the current situation on COVID-19, we are unable to experimentally demonstrate the proposed model. Accordingly we have now limited our proposal to the Discussion section. 

3) Can the authors extend their observations about the role of RAD21 and GRHL2 on PERP and ERBB3 in other cell lines of their collection?

We agree with the reviewer’s comment and our initial intention was to extend our analysis over the ovarian cancer cell line panel. However, for this study we have confined our analysis to SNAI1 overexpression system.

4) The authors should describe the collection and characteristics of SGOCL in a supplementary file.

In the revised version we have removed the data using the SGOCL panel.

5) In the last paragraph of the results section (lines 334-348) the panels of figure 4 have been mislabeled.

In the revised manuscript, we have corrected the mislabelling (Lines 253-275).

6) In line 390 something must be missing in the part “i.e. the intermediate-IE cell lines.”

In the revised manuscript, this part of the discussion was totally revised and therefore needs to correction.

Reviewer 3 Report

The article is addressing an important topic namely the transcriptional regulation of EMT. The authors have a strong background in the field and significant tools to study the details of the investigated scientific question. 

The authors are using a set of cell lines that represent the EMT transition and performed TF KO and expression of EMT-TF-s. 

General remarks: 

The manuscript seems to target a set of researchers with very profound knowledge in the field. The scope of the journal is rather broad therefore the introduction, besides introducing the field, in general, should summarize the works of the group that are the foundation of the present article and are extensively used during the interpretation of the results.

To ease the reading of the article by other scientists, please always introduce the non-trivial abbreviations.  

The font sizes used on Figures 1A, B and 3 A-K are below the accepted limit.

Technical-scientific remarks:

In case of a ChIP protocol,  besides the Ab catalogue numbers please provide the cell number used per IP and the amount of Ab/IP.

The manuscript is relying very much on the assumption that changes in internuclear distance are a measure of changes in apicobasal polarity. The reviewer is sceptical about the validity of this assumption. The previous work of the research group is not proving this assumption. At this stage, this is a hypothesis or speculation.  The fact that in all cases the internuclear distance was changed and the control of these transactions was the empty vector is raising additional concerns. A stronger control would have been a protein that is not involved in there's regulatory processes such as GFP. bGAL, Luc or other inert proteins from the point of human gene expression regulation. 

While the Rad21 changes in expression are interesting and worth to publish the model presented in Figure 5 does not have enough experimental evidence. To claim changes in looping one would need at least classical chromosome conformation capture (3C) experiments. In the case of a small number of genes these experiments are not radically different compared to ChIP, therefore, the lab could probably perform them easily.

In general, the reviewer would suggest providing a more focused manuscript. The chromatin level changes presented together with the changes in expression levels of the presented components of the regulatory system would allow having a message of the manuscript. Some of the results presented in the manuscript or in the supplemental files could be presented in a different manuscript if needed).

The reviewer believes that work is important, reliable and a major revision would allow the publication of the results.

Author Response

Response to Reviewer 3 comments

1) The manuscript seems to target a set of researchers with very profound knowledge in the field. The scope of the journal is rather broad therefore the introduction, besides introducing the field, in general, should summarize the works of the group that are the foundation of the present article and are extensively used during the interpretation of the results.

We agree with the reviewer. In the revised manuscript we have elaborated the introduction section with adequate references.

2) To ease the reading of the article by other scientists, please always introduce the non-trivial abbreviations. 

We have gone over the article, and explained all the abbreviations.

3) The font sizes used on Figures 1A, B and 3 A-K are below the accepted limit.

We have reconstructed Figures 1 and 3 in the revised manuscript.

4) In case of a ChIP protocol, besides the Ab catalogue numbers please provide the cell number used per IP and the amount of Ab/IP.

In the revised manuscript, we have included details about the cell number used per IP and the amount of Ab/IP (Lines 408, 414-415)

5) The manuscript is relying very much on the assumption that changes in internuclear distance are a measure of changes in apicobasal polarity. The reviewer is sceptical about the validity of this assumption. The previous work of the research group is not proving this assumption. At this stage, this is a hypothesis or speculation.  The fact that in all cases the internuclear distance was changed and the control of these transactions was the empty vector is raising additional concerns. A stronger control would have been a protein that is not involved in there's regulatory processes such as GFP. bGAL, Luc or other inert proteins from the point of human gene expression regulation.

We agree with the reviewer’s concern over the assumption of changes in internuclear distance are a measure of changes in apicobasal polarity. In the revised manuscript, we have re-written this section and now described that the increase in internuclear distance of SNAI1-OVCA420 cells as early EMT induced phenotype, whereas increased such increase in SNAI1-OVCA429 cells as complete EMT phenotype (Lines 145-152).

Regarding our control cells, our EV-cells carried stable overexpression of GFP containing pLenti-GIII-CMV-GFP-2A-Puro vector (Line 389).

6) While the Rad21 changes in expression are interesting and worth to publish the model presented in Figure 5 does not have enough experimental evidence. To claim changes in looping one would need at least classical chromosome conformation capture (3C) experiments. In the case of a small number of genes these experiments are not radically different compared to ChIP, therefore, the lab could probably perform them easily.

We agree with the reviewer’s comment on validating the RAD21-mediated regulation of PERP and ERBB3 using 3C experiments. However, due to the COVID-19 situation, our research labs are temporarily closed down, limiting our experimental validation. Therefore, we have restricted our proposed model on chromatin looping regulation to the Discussion section which needs further validation.

7) In general, the reviewer would suggest providing a more focused manuscript. The chromatin level changes presented together with the changes in expression levels of the presented components of the regulatory system would allow having a message of the manuscript. Some of the results presented in the manuscript or in the supplemental files could be presented in a different manuscript if needed).

We agree with the reviewer’s suggestion and accordingly revised the manuscript by removing several data from the main and supplemental figures. We believe the revised manuscript provides better focus on the RAD21 and GRHL2 mediated chromatin regulation.

Round 2

Reviewer 1 Report

The manuscript has been largely reworked with a totally revised approach. This has reduced complexity and eliminated many disturbing elements.. However, the text remains ambiguous about the specific aim of the work. While the title suggests that the main aim of the work was to dissect and characterize the molecular events of the early stages of EMT (in particular the role played by specific DNA binding factors in the early gene expression changes), in the abstract, introduction and discussion, instead, it seems that the aim was to explore the role of the transcription inhibitor Snail, to which great relevance has now been attributed.
The authors should resolve this ambiguity by suitably modifying some parts of the text or title. Furthermore, the whole text should be revised for numerous grammatical and typing errors.

Author Response

The manuscript has been largely reworked with a totally revised approach. This has reduced complexity and eliminated many disturbing elements. However, the text remains ambiguous about the specific aim of the work. While the title suggests that the main aim of the work was to dissect and characterize the molecular events of the early stages of EMT (in particular the role played by specific DNA binding factors in the early gene expression changes), in the abstract, introduction and discussion, instead, it seems that the aim was to explore the role of the transcription inhibitor Snail, to which great relevance has now been attributed.
The authors should resolve this ambiguity by suitably modifying some parts of the text or title. Furthermore, the whole text should be revised for numerous grammatical and typing errors.

We thank the reviewer for acknowledging our revision effort. We understand the reviewer’s concern over the discrepancy between the title and the text. Therefore, we have modified the title in this revision to “SNAI1 driven sequential EMT changes attributed by selective chromatin enrichment of RAD21 and GRHL2”. We believe that this title would better suit the findings described in the text.

Regarding the grammatical and typing errors, we have also gone over the entire manuscript and edited it appropriately (highlighted in gray in the revised manuscript).

Reviewer 2 Report

The revised manuscript by Sundararajan et al is now focused on the involvement of RAD21 and GHRL2 on SNAI1-induced changes associated with EMT. This an improved report that could be strengthened further by the recommended experiments that the authors could not perform. A thorough editing of the revised text is recommended for clarity

Author Response

The revised manuscript by Sundararajan et al is now focused on the involvement of RAD21 and GHRL2 on SNAI1-induced changes associated with EMT. This an improved report that could be strengthened further by the recommended experiments that the authors could not perform. Thorough editing of the revised text is recommended for clarity

We have edited the manuscript appropriately (highlighted in gray in the revised manuscript). We hope now the article is presented with better quality.

Reviewer 3 Report

While the authors have made significant changes in the manuscript, in line with the reports of the other reviewers I consider the conclusions not sufficiently supported by data. The major conclusion that would need experimental confirmation is regarding the looping structure. This part of the manuscript was transferred to the Discussion section, but it is still presented in an affirmative manner, although at this stage it is merely a hypothesis that needs further validation.

For example in the description of Figure 5 the following text can be found:

" GRHL2 and RAD21 modulate transcription of early EMT genes through differential DNA looping along the EMT spectrum. A schematic model illustrating that during epithelial cell state, GRHL2 associates with RAD21 in order to mediate intact chromatin loop structures in ERBB3 (top) and PERP (bottom) loci and thereby enhancing abundant transcription. In intermediate E/M-like cell state, PERP loci showed selective loss of RAD21/GRHL2 enrichment in P12 amplicon region, leading to loss of proximal loop structure, with no significant change in ERBB3 loci. "

Unfortunately, the presented data do not support the differential looping or loss in loop structure, therefore the description is not acceptable.

The words highlighted with red in bold are misleading the reader since they are affirmative.  The whole description of the figure should emphasize that this is "a proposed model" and "based on the results presented a next hypothesis for further research is, that..." etc.

Without careful dissection of what are the conclusions underlined by results and what is mere speculation about possible mechanisms, the manuscript should not be published.

Author Response

While the authors have made significant changes in the manuscript, in line with the reports of the other reviewers I consider the conclusions not sufficiently supported by data. The major conclusion that would need experimental confirmation is regarding the looping structure. This part of the manuscript was transferred to the Discussion section, but it is still presented in an affirmative manner, although at this stage it is merely a hypothesis that needs further validation.

For example in the description of Figure 5 the following text can be found:

" GRHL2 and RAD21 modulate transcription of early EMT genes through differential DNA looping along the EMT spectrum. A schematic model illustrating that during epithelial cell state, GRHL2 associates with RAD21 in order to mediate intact chromatin loop structures in ERBB3 (top) and PERP (bottom) loci and thereby enhancing abundant transcription. In an intermediate E/M-like cell state, PERP loci showed selective loss of RAD21/GRHL2 enrichment in P12 amplicon region, leading to loss of proximal loop structure, with no significant change in ERBB3 loci. "

Unfortunately, the presented data do not support the differential looping or loss in loop structure, therefore the description is not acceptable.

The words highlighted with red in bold are misleading the reader since they are affirmative.  The whole description of the figure should emphasize that this is "a proposed model" and "based on the results presented a next hypothesis for further research is, that..." etc.

Without careful dissection of what are the conclusions underlined by results and what is mere speculation about possible mechanisms, the manuscript should not be published.

We understand the reviewer’s concern over our proposed model and conclusions presented in the manuscript. We have therefore revised the legend of Figure 5 (lines 345-354) and parts of our Discussion section (lines 325-329, 337-343, 355-359, 385-389, highlighted in gray in the revised manuscript). We have addressed all the issues indicated by the reviewer in red and we hope these changes have clearly represented our conclusion over our proposed model.